# Epigenetic Regulation and Post-Translational Modifications of SNAI1 in Cancer Metastasis

**DOI:** 10.3390/ijms222011062

**Published:** 2021-10-14

**Authors:** Bo Dong, Yadi Wu

**Affiliations:** Department of Pharmacology & Nutritional Sciences and Markey Cancer Center, University of Kentucky School of Medicine, Lexington, KY 40506, USA; b.dong@uky.edu

**Keywords:** SNAI1, metastasis, post-translational modifications, epigenetic, EMT

## Abstract

SNAI1, a zinc finger transcription factor, not only acts as the master regulator of epithelial-mesenchymal transition (EMT) but also functions as a driver of cancer progression, including cell invasion, survival, immune regulation, stem cell properties, and metabolic regulation. The regulation of SNAI1 occurs at the transcriptional, translational, and predominant post-translational levels including phosphorylation, acetylation, and ubiquitination. Here, we discuss the regulation and role of SNAI1 in cancer metastasis, with a particular emphasis on epigenetic regulation and post-translational modifications. Understanding how signaling networks integrate with SNAI1 in cancer progression will shed new light on the mechanism of tumor metastasis and help develop novel therapeutic strategies against cancer metastasis.

## 1. Introduction

Tumor metastasis, the spreading of cancer cells from original tumor sites to distant organs followed by development of secondary tumors, is the foremost cause of cancer-related deaths [1]. Initiation of the metastatic program is often followed by exploitation of an embryonic development process referred to as epithelial-mesenchymal transition (EMT) [2]. During EMT, epithelial cells attain mesenchymal phenotypes such as increased motility and invasiveness by dissolving cell–cell junctions and rebuilding cell–matrix connections, accompanied by loss of epithelial markers and a gain of mesenchymal markers [3]. EMT is activated by a plethora of EMT-activating transcription factors (EMT-TFs), such as those from the SNAIL, zinc finger E-box binding homeobox (ZEB), and TWIST families [4].

SNAI1 was the first discovered and most intensively studied transcription repressor of E-cadherin, a hallmark of EMT encoded by the epithelial gene *CDH1*. SNAI1 directly binds to E-boxes present in the *CDH1* promoter to transcriptionally repress its expression. On the other hand, SNAI1 also acts as a transcriptional activator. SNAI1 not only enhances mesenchymal markers including fibronectin, collagens, and the matrix degradation enzyme matrix metalloproteinases 2 and 9 (MMP2 and MMP9), it also increases other EMT transcription factors such as TWIST and ZEB1 [5,6]. In addition, SNAI1 positively regulates transcriptional activation of target genes involved in *Drosophila* development through direct binding to the promoters [7]. In collaboration with early growth response 1(EGR1) and SP1, SNAI1 may directly activate transcription of p15^INK4b^, lymphoid enhancer-binding factor (LEF), and cyclooxygenase 2 (COX2) by directly binding on a consensus motif in HepG2 cells stimulated by the phorbol ester tumor promoter 12-O-tetradecanoyl-phorbol 13-acetate (TPA) [5,8,9,10]. SNAI1 induces resistance to apoptosis, confers tumor recurrence and drug resistance, generates breast cancer stem cell (CSC)-like properties, and induces aerobic glycolysis [11,12,13,14]. Interestingly, SNAI1 is tightly controlled at both transcriptional and protein levels. Many growth factors and cytokines can transcriptionally regulate SNAI1 expression [15]. In addition, SNAI1 protein levels are regulated by post-translational modifications (PTMs). These PTMs have diverse effects on the function of SNAI1.

Because of the reversible plasticity of EMT, epigenetic alternations are required in the EMT process. In eukaryotic cells, genomic DNA interacts with histone proteins and RNA to form chromatin, which holds epigenetic information independent of the DNA genetic data [16]. Alteration of chromatin occurs through regulators responsible for DNA methylation, post-translational modifications of nucleosomal histone tails, and/or non-coding RNA modulation; these epigenetic modifications play a key role in regulating gene expression by defining whether chromatins at a given genomic locus will be transcriptionally active or inactive [17]. For EMT, a variety of epigenetic regulators are critical requirements that interpret signals passed from stimulators to transcription factors [18]. Indeed, the expression of CDH1 is regulated by multiple enzymes involving epigenetic modification. SNAI1 collaborates with multiple epigenetic enzyme complexes, such as DNA methyltransferases, histone deacetylases, and histone methyltransferase and demethylase, in the transcriptional regulation of CDH1. Recent studies suggest a crucial role of epigenetic alterations in the regulation of SNAI1 and EMT markers.

Here, we summarize the regulation of SNAI1 with an emphasis on PTMs. Moreover, we describe recent insights into the epigenetic mechanisms of SNAI1-induced cancer metastasis, focusing on the cooperation of SNAI1 with epigenetic regulators.

## 2. Regulation of SNAI1

Expression of SNAI1 is governed at multiple levels from gene transcription, post-transcriptional regulation, and translation to PTMs such as phosphorylation, ubiquitination, acetylation, and sumoylation.

### 2.1. Structure of SNAI1

SNAI1 belongs to the SNAIL family which consists of SNAI1 (Snail), SNAI2 (Slug), and SNAI3 (Smuc) [19]. The amino termini of SNAI1 contains the evolutionarily conserved SNAI1/Gfi (SNAG) domain, which interacts with several co-repressor complexes or epigenetic remodeling complexes (Figure 1). *Drosophila* SNAIL lacks the SNAG domain but has a consensus PxDLSx motif and exerts their repressive function through the interaction with the co-repressor c-terminal binding protein (CtBP) [19]. In the central region, a serine-rich domain (SRD) is adjacent to the nuclear export sequence (NES) (Scheme 1). SRD controls ubiquitination and proteasome degradation while NES is involved in the regulation of its protein stability and subcellular translocation. The c-terminal zinc finger domain with four C2H2-type zinc fingers is highly conserved. This domain mediates sequence-specific interactions with their target DNA promoters containing an E-box sequence (CAGGTG).

### 2.2. Transcriptional and Post-Transcriptional Regulation

A diverse repertoire of molecular mechanisms regulating SNAI1 at the transcriptional level have been documented in a variety of organisms. Cytokines, chemokines, and growth factors, such as tumor necrosis factor (TNFα), transforming growth factor (TGFβ), interleukin-6 (IL-6), fibroblast growth factors (FGF), epidermal growth factor (EGF), and hepatocyte growth factor (HGF) [20], trigger an intracellular signaling cascade that leads to the binding of a transcription factor to the *SNAI1* promoter to regulate its expression. Extensive evaluation of this regulation has been covered in other excellent reviews [21,22]. Interestingly, the expression of SNAI1 can also be regulated by other EMT-TFs. For example, both SNAI1 and SNAI2 were upregulated under TGFβ stimulation [23]. Depletion of SNAI2 increases SNAI1 expression and vice versa; this compensatory regulation could be indispensable for EMT and cancer progression [24]. In addition, TWIST induces SNAI1 and the Twist-SNAI1 axis is critically involved in EMT and tumor metastasis [25].

Post-transcriptional control provides a fundamental regulatory mechanism for gene expression. Besides regulation by microRNAs [26], SNAI1 transcript stability is also regulated extensively. For example, recent work showed that upon activation of EGF receptor, UDP-glucose 6-dehydrogenase (UGDH) is phosphorylated in human lung cancer cells. Phosphorylated UGDH not only converts UDP-glucose to UDP-glucuronic acid but also interacts with Hu antigen R (an RNA-binding protein that binds to short-lived mRNAs to increase their stability). This interaction attenuates the UDP-glucose-mediated inhibition and therefore enhances the stability of SNAI1 mRNA [27]. In addition, the mRNA of SNAI1 can be modified with N6-Methyladenosine (m6A) by the methyltransferase-like 3 (METTL3) and YTH N6-methyladenosine RNA binding protein 1 (YTHDF1) (m6A readers). m6A in the coding sequence of SNAI1 triggers polysome-mediated translation of SNAI1 mRNA in cancer cells [28]. The stability of SNAI1 mRNA is also enhanced by heterogeneous nuclear ribonucleoprotein, which thus promotes invasion, metastasis, and EMT in breast cancer [29].

### 2.3. Post-Translational Regulation

Because of their critical roles in cancer metastasis, much attention has focused on the PTMs of SNAI1. PTMs function in the regulating protein stability, transcriptional activity, and intracellular localization of SNAI1. Among the number of modifications, phosphorylation and ubiquitination represent the best characterized and control a variety of biological activities, such as apoptosis, transcription, metabolism, and stem cell properties. Therefore, gaining deeper insight into the PTMs may help elucidate important steps in cancer metastasis.

#### 2.3.1. Phosphorylation Regulation

SNAI1 stability is extensively regulated by phosphorylation (Table 1). On one hand, phosphorylation of SNAI1 promotes its proteasomal-mediated ubiquitination degradation. Both casein kinase 1(CK1) and dual specificity tyrosine phosphorylation regulated kinase 2 (DYRK2)-mediated SNAI1 phosphorylation at serine (Ser) 104 act to prime phosphorylations that allow glycogen synthase kinase 3 β (GSK3β)-mediated phosphorylation at Ser96 and Ser100, leading to β-TRCP-induced poly-ubiquitination and degradation [30,31]. Protein kinase D1 (PKD1)-mediated phosphorylation at Ser11 of SNAI1 facilitates F-box protein 11 (FBXO11)-mediated SNAI1 degradation [32]. Under intact apical-basal polarity, α protein kinase C (PKC) kinases promote degradation through phosphorylation of SNAI1 S249 [33]. On the other hand, some SNAI1 phosphorylations prevent its degradation. Most commonly, the main mechanism that regulates SNAI1 stability is phosphorylation at specific sites that reduce its affinity for GSK3β, thus blocking ubiquitination. For example, phosphorylation of SNAI1 at Ser100 by ataxia-telangiectasia mutated (ATM) and DNA-PKCs inhibits SNAI1 ubiquitination by reducing interaction with GSK3β [34,35]. Recently, it was shown that p38 stabilizes SNAI1 through phosphorylation at Ser107, which suppresses DYRK2-mediated Ser104 phosphorylation and subsequent GSK3β-mediated SNAI1 degradation [36]. However, stabilization of SNAI1 also occurs independent of GSK3β. Protein kinase A (PKA) and CK2 have been characterized as the main kinases responsible for in vitro SNAI1 phosphorylation at Ser11 and 92, respectively [37]. Phosphorylation of these two sites control SNAI1 stability and positively regulate SNAI1 repressive function and its interaction with the mSin3A corepressor. Alternatively, confinement of SNAI1 to the nucleus prevents degradation. ERK2-mediated Ser82/Ser104 phosphorylation of SNAI1 leads to nuclear SNAI1 accumulation [38]. P21 (RAC1) activated kinase 1 (PAK1) and GRO-α phosphorylate SNAI1 on Ser246 and increase SNAI1′s accumulation in the nucleus, which thus promotes transcriptional activity of SNAI1 [39,40,41]. Large tumor suppressor kinase 2 (Lats2) phosphorylates SNAI1 at threonine (Thr)203 in the nucleus, which prevents nuclear export, thereby supporting stabilization [42]. Recently, we also found that serine/threonine kinase 39 (STK39) enhances SNAI1 stability by phosphorylation at Thr203 [43]. Notably, phosphorylation can be reversed by phosphatases. We identified c-terminal domain phosphatase (SCP) as a specific phosphatase for SNAI1 [44]. SCP physically interacts with and stabilizes SNAI1 by direct dephosphorylation [44,45].

#### 2.3.2. Ubiquitination and Deubiquitination

SNAI1′s ubiquitination and degradation are controlled by a number of F-box ligases, including β-TRCP1/FBXW1, FBXL14, FBXL5, FBXO11, and FBXO45 [46]. Recently, more E3 ligases have been discovered (Figure 2). F-box E3 ubiquitin ligase FBXO22 elicits antimetastatic effects by targeting SNAI1 ubiquitin-mediated proteasomal degradation in a GSK3β phosphorylation-dependent manner [47]. Through a luciferase-based genome-wide screening using small interfering RNA library against ~200 of E3 ligases and ubiquitin-related genes, SOCS box protein SplA/ryanodine receptor domain and SOCS box containing 3 (SPSB3) was identified as a novel E3 ligase component [48]. SPSB3 targets SNAI1 to promote polyubiquitination and degradation in response to GSK3β phosphorylation of SNAI1. Through yeast two-hybrid screening, the carboxyl terminus of Hsc70-interacting protein (CHIP) was identified as a novel SNAI1 ubiquitin ligase that interacts with SNAI1 to induce ubiquitin-mediated proteasomal degradation [49]. Recently, it was reported that SNAI1 was monoubiquitinated by the ubiquitin-editing enzyme A20. This monoubiquitylation of SNAI1 reduces the affinity of SNAI1 for GSK3β, and thus SNAI1 is stabilized in the nucleus [50].

Deubiquitinases (DUBs) counteract the SNAI1 degradation process to maintain a high level of SNAI1 protein in cancer cells. We recently identified DUB3 as a SNAI1 deubiquitinase that interacts with and stabilizes SNAI1 [51]. Independent research indicated that DUB3 is a target of cyclin-dependent kinase (CDK)4/6, and CDK4/6-mediated activation of DUB3 is essential to deubiquitinate and stabilize SNAI1 [52]. Resistance to platinum-based chemotherapy is a common event associated with tumor dissemination and metastasis in cancer patients. Upon platinum treatment, the ubiquitin-specific protease 1 (USP1) is phosphorylated by ATM and RAD3-related (ATR) and binds to SNAI1. Then, USP1 de-ubiquitinates and stabilizes SNAI1 expression, conferring resistance to platinum, increased stem cell-like features, and metastatic ability [53]. USP29 can be induced by major EMT and metastatic-inducing factors such as TGFβ, TNFα, and hypoxia. This protease enhances the interaction of SNAI1 and SCP1, and results in simultaneous dephosphorylation and de-ubiquitination of SNAI1 and thereafter cooperative prevention of SNAI1 degradation [54]. TGFβ also induces USP27X expression, which increases SNAI1 stability by deubiquitination [55]. Recently, more deubiquitinases have been identified. Eukaryotic translation initiation factor 3 subunit H (EIF3H), OTU deubiquitinase, ubiquitin aldehyde binding 1(OTUB1), USP3, proteasome 26S subunit, Non-ATPase 14 (PSMD14), USP26, USP36, USP37 also target SNAI1 for de-ubiquitination and stabilization (Figure 2) [56,57,58,59,60,61].

#### 2.3.3. Other Post-Translational Regulation

Beyond the well-characterized PTMs of phosphorylation and ubiquitination, at least three other PTMs are involved in regulating SNAI1 protein abundance and activity. First, the sumoylation pathway is very similar to its biochemical analog, ubiquitylation, and regulates diverse cellular processes including transcription and protein stability, chromosome organization, DNA repair, and other cellular processes. TGFβ induces sumoylation of SNAI1 at its lysine (K) 234 residue, which is critical for the EMT-activating function of SNAI1 [62]. Second, the O-linked β-*N*-acetylglucosamine (O-GlcNAc) modification is a monosaccharide addition. SNAI1 is subject to O-GlcNAc at Ser112 under hyperglycemic conditions [63]. This modification leads to stabilization of SNAI1 by inhibition of GSK3β-mediated phosphorylation. Consequently, the O-GlcNAc SNAI1 promotes EMT. Finally, SNAI1 is also acetylated by the histone acetyltransferase adenovirus E1A-associated protein (p300) and CREB binding protein (CBP), two key transcriptional coactivators implicated in a multitude of cellular processes including cancer progression. CBP and p300 interact with SNAI1 to acetylate SNAI1 at K146 and K187, which consequently reduces SNAI1 ubiquitination and thus enhances its protein stability [64] (Figure 3).

## 3. The Interplay between SNAI1 and Epigenetic Regulators in Tumor Metastasis

Because EMT is a reversible and transient process, as well as having reversibility of the epigenetic marks and the enzymatic nature of the regulators, EMT-TFs and chromatin-remodeling enzymes are intimately connected (Figure 3). During tumor metastasis, SNAI1 recruits epigenetic regulators to the *CDH1* promoter, thus repressing its expression. Epigenetic alterations also play a crucial role in SNAI1 expression. The interplay between SNAI1 and epigenetic regulators indicate the complexity of epigenetic mechanisms and the potentially crucial role of histone modifications for regulating SNAI1.

### 3.1. SNAI1 and DNA Methylation

DNA methylation involves a covalent attachment of a methyl group to cytosine residues at CpG-rich dinucleotide sequences through DNA methyltransferases (DNMTs). Upon induction of EMT, hypermethylation of the *CDH1* promoter through DNMTs, which are recruited by EMT-TFs, is constantly observed in a wide variety of cancer cells. For example, SNAI1 interacts with DNMT3A to repress CDH1 expression via DNA hypermethylation and histone modifications of H3K9me2 and H3K27me3 in gastric cancer [65]. Previous research also indicated that DNMT1 was implicated in cell metastasis, such that downregulation or inhibition of DNMT1 could facilitate the metastasis of cancer cells [66]. DNMT1 can decrease the expression of CDH1 by increasing promoter methylation. Interestingly, DNMT1 can also act on CDH1 expression independent of its catalytic activity [67]. DNMT1 interacts with SNAI1 to prevent its interaction with the *CDH1* promoter; this interaction leads to full CDH1 expression. Furthermore, DNMT1 is recruited to the *SNAI1* promoter by AT-rich interactive domain-containing protein 2 (ARID2), a subunit of SWI/SNF chromatin remodeling complex. This complex increases the DNA methylation and suppresses SNAI1 transcription, leading to a repression of EMT. During hepatocellular carcinoma progression, loss/mutation of ARID2 impairs recruitment of DNMT1 to the *SNAI1* promoter. As a result of decreased methylation at the *SNAI1* promoter, there is an upregulation of SNAI1 expression that ultimately promotes EMT [68]. These results suggest that DNMT1 plays a cellular context-dependent role in tumor metastasis. Protein arginine methyltransferase (PRMT) 5 is a type II protein arginine methyltransferase. PRMT5 physically associates with SNAI1 and the NuRD (MAT1) complex to form a transcriptionally repressive complex that catalyzes a simultaneous histone demethylation and deacetylation. In addition, this complex also inhibits tet methylcytosine dioxygenase 1 (TET1) and contributes to DNA hypermethylation [69].

### 3.2. SNAI1 and Histone Modification

#### 3.2.1. Acetylation

A variety of transcriptional co-activating complexes, which contain lysine acetyltransferase, catalyze lysine acetylation of histone tails. Because acetylation masks the positive charge on lysine residues and weakens the DNA–histone association and relaxes the chromatin structure, histone acetylation is often associated with gene activation. SNAI1 recruits the p300 activator complex to the *VEGF* and *Sox2* promoters to stimulate their expression, leading to endothelium generation and tumor growth [70].

#### 3.2.2. Deacetylation

Histone deacetylation by histone deacetylase (HDAC) is believed to restrict gene transcription because it reveals the positive charge of lysine and permits the DNA–histone interaction. HDACs, in particular HDAC1 and HDAC2, are often recruited by EMT-TFs to gene promoter regions and form protein complexes to deacetylate histones and silence expression of epithelial gene factors. For instance, SNAI1 mediates recruitment of the HDAC1/2 that contain Sin3A or NuRD repressor complexes to inhibit CDH1 expression by deacetylation of histones H3 and H4. This effect was abolished by treatment with the HDAC inhibitor Trichostatin A (TSA) [71,72]. Interestingly, HDAC2 can also be recruited by the HOP homeobox to epigenetically inhibit SNAI1 transcription, leading to the enhanced histone H3K9 deacetylation, which subsequently suppresses tumor progression [73]. Similarly, HDAC1 can be recruited by SATB homeobox 2 (SATB2) to the *SNAI1* promoter, repressing *SNAI1* transcription and inhibiting EMT [74]. Recently, it has been reported that HDAC8 increases the protein stability of SNAI1 via AKT/GSK3β signals [75]. HDAC8 interacts with AKT1 to decrease acetylation while increasing its phosphorylation, which further increases Ser9-phosphorylation of GSK3β. Sirt6, the class III histone deacetylates, functions as an NAD^+^-dependent histone deacetylase. Sirt6 interacts with p65 and attenuates NF-kB regulated SNAI1 expression by removing acetyl residues of histone H3K9 and H3K56 in the promoter regions of *SNAI1* [76].

#### 3.2.3. Acetylation Readers

The bromodomain-containing proteins (BRDs) are acetylation readers that bind to ε-*N*-aminoacetyl groups of nucleosomal histone lysines and recruit histone modifiers and transcriptional/remodeling factors to gene promoters; these processes promote upregulation or repression of gene expression. Ever increasing studies in different cancer cells have demonstrated the contribution of BRDs to cancer progression [77]. For instance, BRD4 interacts with SNAI1 if certain K146 and K187 are acetylated. This interaction prevents recognition of SNAI1 by its E3 ubiquitin ligases FBXL14 and β-TrCP1, thereby inhibiting SNAI1 polyubiquitination and proteasomal degradation [78]. In addition, BRD4 increases SNAI1 expression by diminishing the PKD1-mediated proteasome degradation pathway. BRD4 inhibition suppresses the expression of Gli1, which is required for transcriptional activation of SNAI1, indicating that BRD4 controls malignancy of breast cancer cells via both transcriptional and post-translational regulation of SNAI1 [79]. Therefore, inhibition of BRD4 is a promising therapeutic approach for cancer patients with metastatic lesions.

#### 3.2.4. Methylation

Histone lysine methylation is catalyzed by lysine methyltransferases, which directly recruit or inhibit the recruitment of histone-binding proteins. Usually, H3K9 and H3K27 methylation is associated with transcriptional repression, while H3K79 is often linked with gene activation. G9a is responsible for the transcriptionally repressive modification of H3K9. In aggressive lung cancer cells, G9a is preferentially expressed, and its elevated expression correlates with poor prognosis. G9a represses a cell adhesion molecule EPCAM, which stimulates EMT and cancer metastasis by catalyzing H3K9me2 on its promoter [80]. In breast cancer cells, SNAI1 recruits G9a to the *CDH1* promoter for transcription silencing. Therefore, inhibition of G9a reduces promoter H3K9me2 as well as DNA methylation which abrogates EMT and tumor metastasis [81]. Meanwhile, SNAI1 also interacts with Suv39H1, a histone methyltransferase for the trimethylation of histone H3 at lysine K9 (H3K9me3) to the *CDH1* promoter to repress its transcription. EZH2, the catalytic subunit of the polycomb repressive complex 2 (PRC2), promotes transcriptional silencing of CDH1 by H3K27me3 [82,83]. EZH2 can interact with HDAC1/HDAC2 in association with SNAI1 to form a complex that represses CDH1 expression [84]. DOT1L catalyzes the methylation of an active transcription mark histone H3K79, which is crucial for tumor development [85]. In breast cancers, DOT1L forms a transcriptionally active complex with c-Myc and p300 to facilitate H3K79 methylation and acetylation in the promoter regions of *SNAI1* that enhances SNAI1 de-repression, consequently promoting EMT [86].

#### 3.2.5. Demethylation

Histone lysine-specific demethylase 1 (LSD1) functions as an epigenetic regulator by removing methyl groups on the transcription-activating H3K4 or repressing H3K9 residues through an amine oxidase reaction [87,88]. LSD1 takes part in a variety of chromatin-remodeling protein complexes to regulate tumor progression. We found that the amine oxidase domain of LSD1 interacts with the SNAG domain of SNAI1 [89]. SNAI1 recruits LSD1 and forms the SNAI1-LSD1-CoREST complex to repress CDH1 expression and enhance cell migration [89]. Another study indicated that SNAI1 recruits LSD1 on epithelial gene promoters for H3K4me2 demethylation, thereby silencing their expression and promoting EMT [90]. The chromatin remodeling factor Jumonji C (JmjC) domain-containing protein 3 (JMJD3, also known as KDM6B) is a α-ketoglutarate-dependent demethylase which is responsible for the demethylation of di- and trimethyllysine 27 (H3K27m2/3) on histone H3. JMJD3 demethylates H3K27m3 at the *SNAI1* promoter to activate the transcription of SNAI1 during TGFβ-induced EMT [91]. In addition, JMJD1A also transcriptionally activates SNAI1 expression via H3K9me1 and H3K9me2 demethylation at its promoter [92].

## 4. Potential Pharmacological Inhibitors of SNAI1

Given the important role of SNAI1 in driving cancer progression, targeting SNAI1 would be an attractive anticancer therapeutic approach. However, the development of small molecules to inhibit SNAI1′s functions is hindered as there is no clear “ligand-binding domain” for targeting SNAI1. However, other strategies have been successfully attempted. First, the E-box, a SNAI1-binding site, was chosen as a target. A Co(III) complex conjugated to a CAGGTG hexanucleotide was synthesized. This complex binds to SNAI1 and prevents any interaction with DNA, thus reducing the invasive potential of tumor cells [93]. Second, the SNAI1-p53 complex acts as a target. Two leader compounds, GN25 and GN29, increase the expression of p53 and uncouple it from SNAI1. These two compounds selectively inhibit K-ras mutated cells [94]. Third, the LSD1-SNAI1 complex was chosen as a target. Inhibiting its interactions blocks cancer cell invasion [95,96]. Fourth, CYD19, a small-molecule compound, binds to SNAI1 and disrupts the SNAI1 interaction with p300, leading to SNAI1 degradation [97]. CYD19 impairs EMT-associated tumor invasion and metastasis by reversing SNAI1-driven EMT; this finding provides evidence that pharmacologic interference with SNAI1 acetylation may exert potent therapeutic effects in patients with cancer. Finally, chemical classes of synthetic and natural compounds affecting the transcriptional activity and expression of SNAI1 have already been characterized. For example, disulfiram inhibits cell migration, invasion, and growth of tumor grafts through the ERK/NF-κB/SNAI1 signaling pathway [98]. The proteasome inhibitor, NPI-0052, also inhibits SNAI1 expression via inhibition of NF-kB [99]. In all, targeting the SNAI1 complex or suppression of SNAI1 expression is one major approach to specifically inhibit SNAI1 activity.

Proteolysis-targeting chimeras (PROTACs) that hijack the ubiquitin-proteasome system for targeted protein degradation have expanded significantly in years [100]. This technology circumvents some of the limitations associated with traditional small-molecule therapeutics. PROTAC consists of a ligand for an E3 ligase and a ligand for a protein of interest (POI) connected by a chemical linker to form a ternary complex. In 2021, TRAnscription Factor Targeting Chimeras (TRAFTACs) technology was developed. The TRAFTAC system is composed of a HaloTag-fused dCas9 protein and a chimeric oligonucleotide that can bind transcription factor of interest (TOI) and dCas9 simultaneously [101]. This system labels the TOI with ubiquitin which then degrades the TOI by proteasomal machinery. This strategy was applied to target several transcription factors including E2F1 and NF-kB [102]. It will be attractive to design a TRAFTAC targeting SNAI1.

## 5. Conclusions and Perspective

SNAI1 as the key EMT regulator plays important roles in invasion and metastasis. The molecular events mediated by SNAI1 are of interest as therapeutic targets, in particular for resistant metastatic tumors. Although direct targeting of SNAI1 is unsuccessful, identifying inhibitors for PTMs of SNAI1 hold significant potential, and thus are a high priority in the development of future cancer treatments. Indeed, many pharmacological approaches, including chemical inhibitors and monoclonal antibodies that target these modification enzymes including deubiquitinase and kinase, have been devised and show promise for the treatment of tumor metastasis [103]. Furthermore, identification of SNAI1′s post-transcriptional and PTMs is crucial given that these changes could be identified in the primary tumor before metastasis occurs. Such knowledge would facilitate better prediction of patients who have genotypes that are more likely to follow an aggressive clinical course and who are prone to development of metastases.

In addition, because of the intimate connection between SNAI1 and chromatin-remodeling enzymes, targeting the epigenetic enzymes to reverse the EMT process is also an efficient and promising approach [104]. Indeed, abundant pre-clinical and clinical studies examining the effects of these epigenetic enzyme inhibitors alone or in combination with other anti-cancer agents are under development [105]. However, the impact of these epigenetic alternations on tumor metastasis differs greatly in various types of cancers. Therefore, it is urgent to comprehensively understand the mechanisms of action and roles of epigenetic modulations on EMT in different cancer types. These detailed mechanisms of epigenetic regulation in tumor metastasis will provide a bright future for the use of an efficient and specific “epigendrug” as one of the important therapeutic strategies in the fight against tumor metastasis.

## Figures and Tables

**Figure 1 ijms-22-11062-f001:**
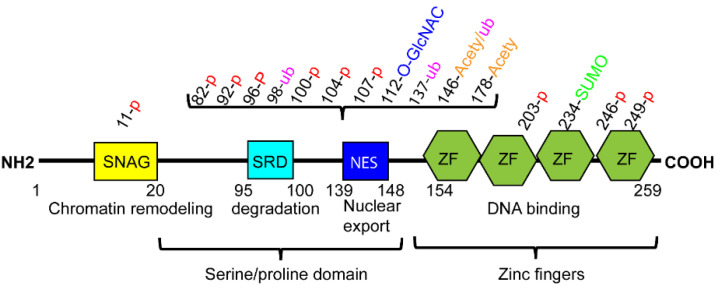
The structure of SNAI1 and potential post-translational modification sites. SNAI1 contains four main domains: Scheme 1. Gfi-1 (SNAG) domain, serine-rich domain (SRD), nuclear export sequence (NES) and zinc-finger (ZF) domain. P: phosphorylation. Acety: Acetylation. Ub: Ubiquitination. SUMO: Sumoylation.

**Figure 2 ijms-22-11062-f002:**
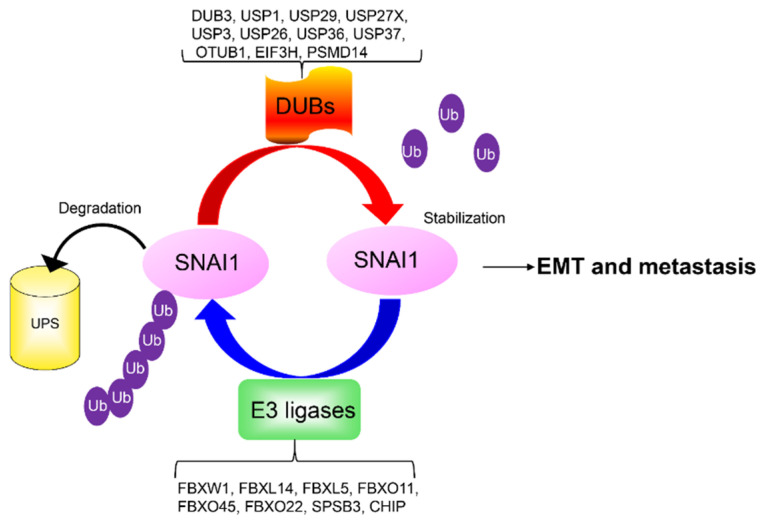
The ubiquitination and de-ubiquitination of SNAI1. SNAI1 is degraded by multiple E3 ligases. By contrast, de-ubiquitinases counteract E3 ligase activity and prevent SNAI1 degradation. USP: ubiquitin-specific protease; OTUB1: OTU deubiquitinase, ubiquitin aldehyde binding 1; PSMD14: proteasome 26S subunit, Non-ATPase 14; EIF3H: eukaryotic translation initiation factor 3 subunit H; DUB: deubiquitinase; UPS: ubiquitin/proteasome system; FBXW1: F-box/WD repeat-containing protein 1; FBXL: F-box and leucine rich repeat protein; FBXO: F-boxes other; SPSB3: SplA/ryanodine receptor domain and SOCS box containing 3; CHIP: carboxy-terminus of Hsc70 interacting protein.

**Figure 3 ijms-22-11062-f003:**
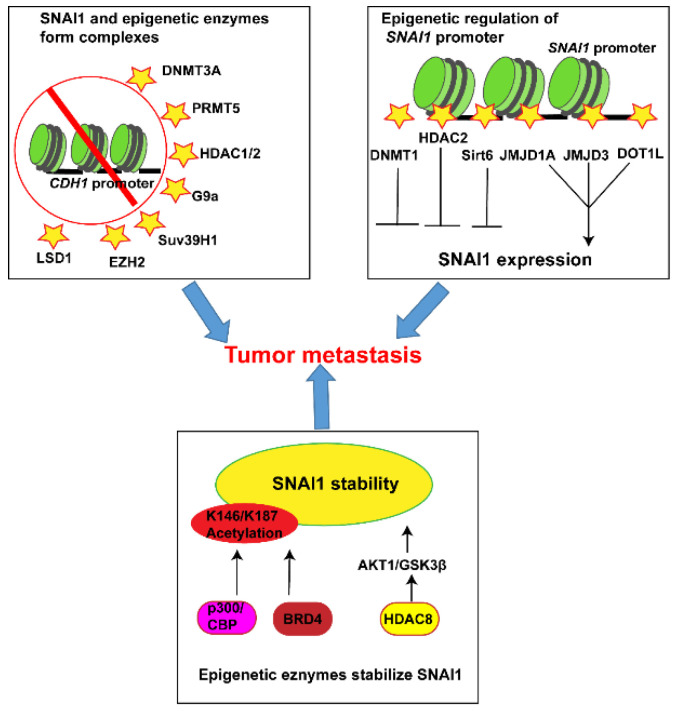
The interplay between SNAI1 and epigenetic regulators. SNAI1 collaborates with epigenetic regulators to repress CDH1 expression. Epigenetic regulators are recruited to the *SNAI1* promoter, leading to transcriptional activation or repression of SNAI1. In addition, epigenetic regulators regulate SNAI1 stability by post-translational modifications. The increase in SNAI1 expression via multiple epigenetic mechanisms leads to the cancer metastasis that is accompanied by the loss of CDH1. DNMT: DNA methyltransferases; PRMT: protein arginine methyltransferases; HDAC: histone deacetylases; Suv39H1: suppressor of variegation 3-9 homolog 1; EZH2: enhancer of zeste 2 polycomb repressive complex 2 subunit; LSD1: lysine-specific demethylase 1; JMJD: Jumonji C domain-containing; DOT1L: DOT1 like histone lysine methyltransferase; BRD4: bromodomain-containing protein 4; CBP: CREB binding protein.

**Table 1 ijms-22-11062-t001:** Phosphorylation regulators involved in SNAI1.

Function	Regulation Factor	Phosphorylation Sites
Degradation	PKD1	Ser11
	αPKC	Ser249
	CK1	Ser104/Ser107
	GSK3β	Ser96/Ser100/S104/Ser107
	DYRK2	Ser104
Stabilization	PTK6	Tyr342
	ATM	Ser100
	CK2	Ser92
	p38	Ser107
Nuclear accumulation and stabilization	ERK	Ser82/Ser104
	PAK1	Ser246
	GROα	
	LATS2	Thr203
	STK39	

## Data Availability

Not applicable.

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
