# Peer review of "Epigenetic Regulation and Post-Translational Modifications of SNAI1 in Cancer Metastasis"

_ijms, 2021, doi:10.3390/ijms222011062_

Round 1
Reviewer 1 Report
The review by Dong and Wu covers our current knowledge of the mechanisms involved in the control of SNAI1 levels and activities. Relevance towards the pharmacological targeting of SNAI1 for cancer treatment is discussed.
I think that the review should be improved along the following lines:
- The authors only briefly allude to other pro-EMT TFs. How SNAI1 expression/activities relates to that of additional well established pro-EMT TFs should be further presented. This should include other SNAI family members and potentially other E-box recruited TFs.
- Interactions between SNAI1 and epigenetic regulators is a central point of this review. Fig3 establishes that interaction with chromatin remodeler complexes can regulate SNAI1’s levels or activities at different levels. Control of SNAI1 stability is actually already discussed in section2. In section3, different types of chromatin remodeling complexes are listed and how they relate to SNAI1 is described, including regulation of SNAI1 gene or SNAI1-mediated gene regulation together with regulation of SNAI1 stability (i.e. non-histone/epigenetic activities). In the end, this organization appears confusing.
- Could the authors discuss whether SNAI1 should only be considered as a repressor or whether it can or could potentially act as a transcriptional activator. For instance, SNAI1 interacts with CBP/P300 and BRD4, which are commonly described as involved in transcriptional activation.
- When discussing transcriptional regulation mediated by SNAI1, the authors only refer to the CDH1 gene. Could the authors extend the discussion to indicate what is known and still unknown regarding SNAI1 genome-wide control of gene expression beyond CDH1.
- Section 4 is a topic of interest and should be extended to better explain current and potential future strategies which may allow to target SNAI1 in cancer.
Author Response
The authors only briefly allude to other pro-EMT TFs. How SNAI1 expression/activities relates to that of additional well established pro-EMT TFs should be further presented. This should include other SNAI family members and potentially other E-box recruited TFs.
Responses: Thanks reviewer for great suggestions. It is important to present coordinated functions among these EMT TFs. We added this information from line 34 to line 35. We also added a section on co-regulation between SNAI1 and EMT TFs in section 2.2 from line 101 to line 105.
- Interactions between SNAI1 and epigenetic regulators is a central point of this review. Fig3 establishes that interaction with chromatin remodeler complexes can regulate SNAI1’s levels or activities at different levels. Control of SNAI1 stability is actually already discussed in section2. In section3, different types of chromatin remodeling complexes are listed and how they relate to SNAI1 is described, including regulation of SNAI1 gene or SNAI1-mediated gene regulation together with regulation of SNAI1 stability (i.e. non-histone/epigenetic activities). In the end, this organization appears confusing.
Responses: We moved the acetylation from section 3.2.1 to section 2.3.3.
- Could the authors discuss whether SNAI1 should only be considered as a repressor or whether it can or could potentially act as a transcriptional activator. For instance, SNAI1 interacts with CBP/P300 and BRD4, which are commonly described as involved in transcriptional activation.
Responses: Thanks reviewer for these constructive suggestions. SNAI1 not only functions as a repressor but also acts as activator. We added this information to the Introduction and Section 3.2.1.
- When discussing transcriptional regulation mediated by SNAI1, the authors only refer to the CDH1 gene. Could the authors extend the discussion to indicate what is known and still unknown regarding SNAI1 genome-wide control of gene expression beyond CDH1.
Responses: We extended the discussion about the SNAI1 targeting. This information was added in the Introduction and Section 3.2.1.
- Section 4 is a topic of interest and should be extended to better explain current and potential future strategies which may allow to target SNAI1 in cancer.
Responses: We extended the discussion of SNAI1 inhibitors. We not only discussed more inhibitors but also provide potential future strategies that focused on PROTAC, particular the TRAFTACs system.
Reviewer 2 Report
When going down into the molecular mechanisms of cancer development and progression the number of possible causes increases. The role of SNAI1 as a member of SNAIL/SNUG family has been recognized over 20 years ago. It is involved in the process om EMT of tumor cells and changes in e -cadherin function in metastasis formation. This paper gathers multiple papers on epigenetic and post-translational regulations of SNAI1 in cancer metastasis.
It is neither a new concept nor a revolutionary idea but the review of possible models of SNAI1 regulation is comprehensive and up to date.
There is still no evidence of highly possible therapeutic use of these findings. Some conclusions of this paper go to far.
In general there is no formal comments, this paper is well written and up to date but the importance is unknown.
Author Response
Thanks a lot for reviewer's comments. We really appreciate it. In this review, we summarize the PTM and epigenetic regulation of Snail during tumor metastasis. Hope these mechanism will help develop novel therapeutic strategies against caner metastasis.
Round 2
Reviewer 1 Report
The authors have appropriately improved the manuscript based on my recommendations